# Serum Levels of α-Klotho, Inflammation-Related Cytokines, and Mortality in Hemodialysis Patients

**DOI:** 10.3390/jcm11216518

**Published:** 2022-11-02

**Authors:** Katarzyna Aleksandra Lisowska, Hanna Storoniak, Monika Soroczyńska-Cybula, Mateusz Maziewski, Alicja Dębska-Ślizień

**Affiliations:** 1Department of Pathophysiology, Faculty of Medicine, Medical University of Gdańsk, 80-211 Gdańsk, Poland; 2Department of Nephrology, Transplantology and Internal Medicine, Faculty of Medicine, Medical University of Gdańsk, 80-214 Gdańsk, Poland

**Keywords:** α-Klotho, cytokines, hemodialysis, cardiovascular events, mortality

## Abstract

It has been hypothesized that α-Klotho deficiency might contribute to chronic inflammation in patients with end-stage renal disease (ESRD), especially those on hemodialysis (HD). Serum Klotho levels by some authors are considered a potential predictor of cerebrovascular events. Therefore, we analyzed serum levels of α-Klotho with ELISA and inflammation-related cytokines in HD patients. Sixty-seven HD patients and 19 healthy people were recruited between November 2017 and June 2021. A Cytometric Bead Array (CBA) was used to determine the level of different cytokines: IL-12p70, TNF, IL-10, IL-6, IL-1β, and IL-8. A human Klotho ELISA kit was used to determine the level of α-Klotho in the plasma samples of HD patients. There was no difference in serum levels of α-Klotho between HD patients and healthy people. Patients had increased serum IL-6 and IL-8. Significant positive correlations existed between the concentration of α-Klotho and the serum concentrations of IL-12p70, IL-10, and IL-1β. However, in a multivariable linear regression analysis, only patients’ age was associated independently with α-Klotho level. Serum α-Klotho was not associated with higher mortality risk in HD patients. While these results draw attention to potential relationships between α-Klotho proteins and inflammatory markers in HD patients, our cross-sectional study could not confirm the pathogenic link between α-Klotho, inflammation, and cardiovascular mortality.

## 1. Introduction

The *KLOTHO* gene encoding the protein of the same name, discovered at the end of the 20th century, is one of the genes affecting animal and human life span [1]. Its knock-out mutations in mice induced osteoporosis, vascular calcification, muscle atrophy, hypoglycemia, and hyperphosphatemia and promoted reduced activity, and hearing impairment, generally leading to accelerated aging [2]. Many of the animal studies were later partially confirmed in humans. One of the best-known functions of Klotho, which belongs to the β-glucuronidase family, is its role in regulating calcium and phosphate metabolism, among others, by regulating pathways dependent on fibroblast growth factor 23 (FGF23) [3]. The kidneys are one of the organs in which Klotho expression is highest [4].

We know about three Klotho isoforms, of which α-Klotho is best known. It exists either in a form associated with the cell membrane or a secretory form (sKlotho). Secretive Klotho may arise as a result of the extracellular part being excised from the cell membrane or alternative splicing [5]. One of the essential functions of α-Klotho (both membrane and secretory) is the role of the FGF23 cofactor in increasing renal phosphate secretion [6]. sKlotho, with circulation, also goes to various tissues, inhibiting insulin-like growth factor (IGF) dependent pathways [7].

Chronic kidney disease (CKD) is associated with the loss of active nephrons in the renal parenchyma and inevitably leads to end-stage renal disease (ESRD), requiring renal replacement therapy. In addition, the course of the disease is complicated by cardiovascular disease, anemia, malnutrition, acid-base, and calcium-phosphate disorders. Furthermore, the loss of active renal parenchyma leads to hypocalcemia, hyperphosphataemia, and renal osteodystrophy, which are associated with a gradual decrease in serum sKlotho, as shown in a study by Kim et al. [8]. According to the authors, among CKD patients with low α-Klotho concentrations (below the median value of 280 pg/mL), a doubling of creatinine levels within a couple of months of observation occurs in a higher percentage than in the patients with concentrations above 280 pg/mL. α-Klotho concentration also positively correlates with the estimated glomerular filtration rate (eGFR) and calcium level [8].

In hemodialysis (HD) patients, α-Klotho levels are also associated with cardiovascular morbidity and mortality [9,10]. Otani-Takei et al. [10] showed that HD patients with low Klotho serum concentrations (<309 pg/mL) are more likely to die from cardiovascular causes. Meanwhile, HD patients with high concentrations of sKlotho have reduced occurrence of outcomes combining cardiovascular events and cardiovascular death [11]. Therefore, serum Klotho level by some authors is considered a potential predictor of cerebrovascular disease in HD patients. 

ESRD patients, especially those on hemodialysis, are characterized by increased oxidative stress and inflammation, which seem to be central components of the uraemic phenotype. Chronic inflammation has been linked with cardiovascular mortality [12] and protein-energy wasting (PEW) in HD patients [13]. Inflammatory markers, especially interleukin (IL)-6, are also strong predictors of poor outcomes in ESRD patients [14]. The pathophysiology of chronic inflammation in CKD is a consequence of multiple factors, including the type of dialysis membrane, uraemic toxins, oxidative stress, cellular senescence, gut dysbiosis, hypoxia, and fluid and sodium overload [15]. It has been hypothesized that α-Klotho deficiency might contribute to increased oxidative stress and inflammation in ESRD patients. Low concentrations of α-Klotho are associated with inflammation in experimental kidney disease models [16]. Human studies show that in patients on peritoneal dialysis (PD), high concentrations of Klotho (above the median value of 329 pg/mL) are correlated with lower IL-6 levels [17]. In HD patients, the problem of the relationship between α-Klotho and inflammation is still poorly understood. Therefore, this work aimed to analyze correlations between the serum concentration of α-Klotho and cytokines associated with inflammation (IL12p70, TNF, IL-10, IL-6, IL-1β, IL-8) in HD patients. In the present study, we also investigated the predictive significance of serum levels of α-Klotho and inflammation-related cytokines in terms of survival and cardiovascular events.

## 2. Material and Methods

### 2.1. Patients

HD patients were recruited between November 2017 and June 2021. The study group comprised 67 HD patients (Table 1). Nineteen healthy people were also recruited. The Independent Bioethics Committee for Scientific Research at the Medical University of Gdańsk approved the study. We performed all the experiments following the relevant guidelines and regulations. All participants were informed about the purpose of the tests and gave written informed consent. A nephrologist thoroughly reviewed each patient’s medical chart, and data were extracted on the primary cause of CKD, prescribed medications, presence of cardiovascular disease (CVD), and other comorbid conditions, such as diabetes mellitus (DM). Laboratory data included serum concentrations of creatinine, parathormone (PTH), calcium (Ca), and phosphorus (Pi). eGFR was calculated using the CKD-EPI equation according to the KDIGO guidelines.

All HD patients had an estimated glomerular filtration rate (eGFR) below 15 min/mL/1.73 m^2^ and underwent four-hour hemodialysis sessions three times a week using low-flux NIPRO PES 150DL, 170DL, 210DL, or high-flux ELISIO 15H and 17H dialyzers. In addition, patients had regular dialysis adequacy assessment by measuring urea clearance using the equation Kt/V (K—urea clearance, t—time on dialysis, V—volume of distribution). Healthy people underwent a routine health examination, had no history of medical disease, and were not taking regular medication.

In 13 patients, the primary cause of chronic kidney disease was glomerulonephritis (GN), in 25—diabetic nephropathy (DN), in 13—ischemic nephropathy (IN), in 4—hypertensive nephropathy (HN), in 6—adult polycystic kidney disease (ADPKD). In two patients, the primary cause of CKD was unknown. Two patients had obstructive nephropathy, and two had granulomatosis with polyangiitis (GPA). None of the patients suffered from active infection, inflammation, malnutrition, malignancy, or blood loss during the study. They did not receive steroids or immunosuppressive agents. 

Blood samples were collected from each patient before the HD session into gel-separator serum tubes to collect serum to assess concentrations of cytokines and α-Klotho. We stored serum samples at −80 °C as recommended by other authors [18,19].

### 2.2. Cytokine Measurement in Plasma Samples

BD™ Cytometric Bead Array (CBA) Human Inflammatory Cytokines Kit (Becton, Dickinson and Company, Franklin Lakes, NJ, USA) was used according to the manufacturer’s protocol to determine the level of different cytokines, i.e., IL-12p70, TNF, IL-10, IL-6, IL-1β, and IL-8, in the plasma samples of HD patients. The detection range for all measured cytokines was between 20 and 5000 pg/mL. We performed quantitative cytometric fluorescence analysis with the FACScan cytometer (Becton, Dickinson and Company, Franklin Lakes, NJ, USA). Cytokine concentrations were analyzed with the use of BD™ CBA software (Becton, Dickinson and Company, Franklin Lakes, NJ, USA).

### 2.3. Detection of α-Klotho Alpha in Plasma Samples

Human KL(Klotho) ELISA Kit (Wuhan Fine Biotech Co., Wuhan, China) was used according to the manufacturer’s protocol to determine the level of α-Klotho in the plasma samples of HD patients. The effective measurement range of this assay was between 7.8 and 500 pg/mL. The density of the yellow color was proportional to the target amount of sample captured in the 96-well plate, and the O.D. absorbance was read at 450 nm absorbance in Epoch™ Microplate Spectrophotometer (Agilent Technologies, Inc., Santa Clara, CA, USA).

### 2.4. Outcome Measurements

The primary outcomes included all-cause mortality, cardiovascular mortality, or a combined outcome that included cardiovascular death or non-fatal cardiovascular events, defined as myocardial infarction (MI), stroke, major arterial/venous thrombotic episode, coronary revascularization, atrial fibrillation (AF), or decompensated heart failure (HF). Cardiovascular death was defined as death from any cardiovascular event. Cardiovascular events and cause of death were assessed by the attending physicians, who were unaware of serum α-Klotho. Censoring was performed either on the date of the first occurrence of each studied endpoint or on 30 June 2022. The patient’s usual medications, including antihypertensive agents, erythropoiesis-stimulating agents (ESAs), or alfacalcidol, were continued during the observation period.

### 2.5. Analysis and Statistics 

The concentration of α-Klotho was calculated with GraphPad Prism 9 (GraphPad Software Inc., San Diego, CA, USA). GraphPad Prism 9 was also used to perform statistical analyses. The Kolmogorov-Smirnov and Lilliefors tests were used for testing normality. Significance tests were chosen according to data distribution with a significance level of *p* < 0.05. Comparisons between the two groups were assessed with the nonparametric Mann-Whitney U test for continuous variables. Correlations between continuous variables were done using the Spearman Rank Correlation test. Cox regression models were used to analyze the relationships between the primary outcomes and the serum levels of α-Klotho and cytokines in HD patients. Data are expressed as medians with minimum and maximum results, percentages, or hazard ratios with 95% confidence intervals (95% CI).

## 3. Results

### 3.1. Baseline Demographic, Clinical, and Laboratory Findings

The clinical information about 67 HD patients, including serum concentrations of creatinine, PTH, Ca, Pi, levels of cytokines IL-12p70, TNF, IL-10, IL-6, IL-1β, IL-8, and α-Klotho are shown as mean and median values in Table 1. Fifty-eight patients (87%) patients were diagnosed with chronic heart failure (CHF). Sixty-three (94%) patients were suffering from hypertension. In addition, 36 (55%) patients were diagnosed with DM. In 25 patients, DN was a primary cause of CKD. 36 (54%) of patients received alfacalcidol 0.25 µg daily.

There was no significant difference in the serum concentration of α-Klotho between healthy people and HD patients. However, HD patients had higher IL-6 and IL-8 concentrations than healthy people (Table 1). 

The primary cause of CKD or the type of DM did not influence the serum concentration of α-Klotho (data not shown).

### 3.2. Factors Associated with α-Klotho Levels

Bivariate correlation analyses revealed that there was a significant negative correlation between the concentration of serum α-Klotho and patients’ age (r = −0.268, *p* = 0.028) and the serum concentration of PTH (r = −0.290, *p* = 0.017) (Table 2). No correlation has been found between α-Klotho and Ca or Pi. The serum concentration of α-Klotho was positively correlated with serum IL-12p70 (r = 0.250, *p* = 0.041), IL-10 (r = 0.333, *p* = 0.006), and IL-1β (r = 0.436, *p* < 0.001). A multiple linear regression analysis was conducted to identify further factors independently associated with Klotho. Age, PTH, IL-12p70, and IL-10, which were significantly correlated with α-Klotho in bivariate correlation analyses, were used for the analysis. Only age was associated independently with patients’ Klotho levels (Table 3).

As for other relationships between inflammatory markers and clinical parameters, the concentration of IL-8 was negatively correlated with HD time (r = −0.278, *p* = 0.023) (Table 2). Creatinine, PTH, and Pi levels were negatively correlated with patients’ age. Other correlations are shown in Table 2.

### 3.3. Factors Associated with Cardiovascular Events and Mortality

Fourteen patients died within the median follow-up period of 7 months. Nine patients died because of cardiovascular events. Three patients developed an infection and died because of sepsis. Two patients died because of unknown reasons and died out of the hospital. The Cox regression analyses revealed several factors associated with cardiovascular events, cardiovascular mortality, and all-cause mortality (Table 4). Serum creatinine levels were predictive for cardiovascular events (HR 0.726, 95% CI 0.575–0.902, *p* = 0.005), cardiovascular mortality (HR 0.517, 95% CI 0.313–0.807, *p* = 0.006), and all-cause mortality (HR 0.577, 95% CI 0.392–0.816, *p* = 0.003). eGFR was predictive for cardiovascular death (HR 1.25, 95% CI 1.07–1.46, *p* = 0.003) and all-cause mortality (HR 1.20, 95% CI 1.05–1.36, *p* = 0.003). DN was predictive for cardiovascular events (HR 2.73, 95% CI 1.31–6.11, *p* = 0.009).

## 4. Discussion

The history of Klotho is associated with discovering the relationship between its expression and aging in mice [1]. Among the three isoforms, the most important is α-Klotho, which occurs in the kidney in high concentrations. It is produced mainly in the distal nephron tubules and is associated with calcium-phosphate metabolism [3]. Over the years, many publications have emerged showing significant correlations between the concentration of α-Klotho in the blood of CKD patients and the progression of the disease [8] and cardiovascular complications, especially in HD patients [9,10]. Meanwhile, the relationship between α-Klotho and inflammatory markers in HD patients is still unclear. Therefore, our study compared serum levels of selected cytokines and α-Klotho between HD patients and healthy people of similar age.

Some studies show that HD patients have high serum IL-1, IL-6, IL-8, and TNF-α [20], produced mainly by mononuclear cells [21]. According to some authors, IL-6, in particular, is recognized as a strong predictor of poor outcomes in ESRD patients [14]. In our study, HD patients had higher IL-6 and IL-8 concentrations than healthy people but not IL-1β or TNF-α. In addition, there was no difference between patients and control in serum levels of α-Klotho. Until recently, studies showed a significant positive correlation between soluble α-Klotho and eGFR in patients with CKD [8,22]. According to some authors, α-Klotho expression is also decreased in other tissues, including parathyroid glands [23] or vascular smooth muscles [24]. However, not all authors agree on whether α-Klotho’s serum levels in HD patients reflect its tissue deficiencies. For example, Seiler et al. [25] analyzed plasma levels of Klotho in a large cohort of patients in CKD stages 2–4 and found no decline as the disease progressed. The authors saw no correlation between Klotho levels and eGFR or parameters associated with calcium-phosphate metabolism. Yildirim et al. [26] have demonstrated that HD and PD patients have higher serum α-Klotho than healthy people. In a recent study by Gamrot et al. [27], mean serum α-Klotho was higher in children with CKD on dialysis than in conservatively treated children. Drüeke et al. [28] suggested that there might be a difference in Klotho levels between patients supplemented with vitamin D and those not receiving vitamin D. In our study, approximately half of the patients were supplemented with alfacalcidol. However, it did not influence α-Klotho serum concentrations. As in the study by Seiler et al. [25], the α-Klotho level was negatively correlated with HD patients’ age, which was confirmed by multiple linear regression analysis. The correlation matrix showed no relationships between serum α-Klotho and any variables associated with CKD, including creatinine, eGFR, or calcium and phosphorus concentrations, except for PTH. Similar results were reported by Wei et al. [11], who also saw no correlation between serum Klotho and Ca, Pi, or PTH in HD patients despite a much larger study group. Buiten et al. [29] demonstrated a correlation between sKlotho only with PTH in HD patients.

Seiler et al. [25] stressed in their study that a reliable measurement of circulating Klotho has become available only recently, with more sensitive and specific ELISA tests. Therefore, we must consider that changes in the serum levels of α-Klotho in HD patients are not necessarily directly related to disease progression. It is also possible that in HD patients, the eGFR is already too low and the calcium-phosphate balance too disturbed to demonstrate any relationship with Klotho levels. Since there is no relationship between serum α-Klotho and calcium-phosphate balance, we can assume that the clinical impact of soluble Klotho might not be the same as tissue Klotho.

In the latest studies, soluble α-Klotho has been shown to be significantly associated with well-recognized inflammatory biomarkers, e.g., uric acid (UA) [30,31], C-reactive protein (CRP), white blood cell (WBC) count, and mean platelet volume (MPV) [31]. In PD patients, high concentrations of Klotho (above the median value of 329 pg/mL) were associated with lower IL-6 levels [17]. In children with CKD, no correlation was observed between serum α-Klotho or TNF-α concentration and any measured anthropometric and laboratory parameters [27]. In our study, α-Klotho also positively correlated with IL-12p70, IL-10, and IL-1β but not with TNF or IL-6. Downregulation of renal α-Klotho expression has been shown to increase kidney inflammation [16] and induce renal fibrosis [32] in animal models. In mice that suffered unilateral renal ischemia and reperfusion injury (IRI), recombinant Klotho prevented an increase in serum IL-6, IL-1β, and TNF-α levels [33]. These processes are related to the role of Klotho in reducing the production of reactive oxygen species (ROS) and reactive nitrogen species (RNS) [34], suppressed TNF-induced NF-kappaB activation [35], and upregulated anti-inflammatory IL-10 [34]. Thus, the anti-inflammatory role of α-Klotho could explain a positive correlation with anti-inflammatory IL-10 in our study group. However, till now, such an association was only confirmed in patients with established cardiovascular disease (CVD), whose low Klotho concentrations accompanied low IL-10 levels [36]. 

However, what is surprising is the positive correlation between α-Klotho and IL-1β and IL-12p70, which have a pro-inflammatory effect. IL-12 is a pro-inflammatory cytokine that induces the production of interferon-γ (IFN-γ) by T helper 1 (T_h_1) cells associated with cell-mediated immunity. Studies showed that HD patients exhibit an increased percentage of T_h_1 cells compared with healthy controls, and their monocytes produce high levels of IL-12 [37]. IL-1β, also known as leukocytic pyrogen, is an essential mediator of the inflammatory response associated with acute symptoms such as fever and hypotension in HD patients [38]. Both cytokines are associated with attenuated inflammatory responses in ESRD, aggravated by high uremic toxins and repeated contact with the dialysis membrane [20]. Therefore, their positive correlation with α-Klotho draws attention to its potential, not yet described, role in promoting inflammation. The relationship between IL-10 and IL-1β levels is most likely the result of monocytes and macrophages responding to high levels of pro-inflammatory cytokines. In particular, macrophages produce IL-10 in a negative feedback loop to reduce uncontrolled inflammatory cytokine production during, for example, infection [39]. By binding with its receptor on cells of innate immunity, IL-10 inhibits the release of pro-inflammatory cytokines and decreases antigen presentation and phagocytosis. Therefore, it would be expected that the higher IL-10 level would accompany high IL-1 levels. In our study, those two variables were correlated.

Several articles have reported a significant association between serum Klotho and mortality in HD patients [9,10,11,40]. In our study, serum α-Klotho levels did not influence cardiovascular events or all-cause mortality in HD patients. In physiologic conditions, healthy kidneys maintain soluble Klotho levels [41]. However, as mentioned earlier, other organs may participate in maintaining soluble α-Klotho in ESRD, and high α-Klotho expression has been found in the parathyroid gland [23] and vascular smooth muscles [24]. In animals, high tissue α-Klotho has also been reported in the choroid plexus [42], while in humans it has been detected in cerebrospinal fluid [43]. While there is one *KLOTHO* gene, three isoforms of α-Klotho exist, the transmembrane form, a shed soluble form, and a truncated soluble form produced by alternative splicing. Its soluble form results from direct secretion by the cell of truncated form or from cleavage of the extracellular domain of the full-length protein [5]. The different tissue locations of α-Klotho and the different mechanisms of soluble Klotho production bring into question the relationship between protein expression in the kidney and its amount in the blood in CKD patients [11,25,26,27]. The lack of relationship between serum α-Klotho and fatal or non-fatal cardiovascular events could be explained by several factors. First, CVD starts to develop in the early stages of CKD. Second, many factors are associated with CVD progress, including smoking, obesity, diabetes, or hypertension. Third, serum α-Klotho simply may have no independent influence on CVD development in HD patients. Several authors have obtained similar results. For example, Buiten et al. [29] reported that serum sKlotho in patients in CKD stage 5D was not independently associated with cardiovascular disease. Moreover, HD patients with a low sKlotho level (<460 pg/mL) did show coronary artery disease (CAD) and left ventricular (LV) dysfunction more frequently. Seiler et al. [25] also demonstrated that sKlotho did not predict any cardiovascular outcomes, including death.

At present, we do not know what the “normal” serum level of α-Klotho is. Presently, there is no standardized assay to measure circulating Klotho in humans. It is also possible that the distinction and characterization of the soluble forms of α-Klotho may be necessary to fully unveil the role of Klotho in CKD. Many authors have used median splits to demonstrate its clinical relevance in CKD or CVD. However, the problem with median splits is that when a continuous variable is categorized, every value above the median, for example, is considered equal. Hence, it is not surprising that the results from different centers regarding the relationship between serum Klotho levels and clinical parameters in dialysis patients do not coincide.

The risk of CVD in CKD patients is far greater than in the general population, with CVD mortality in HD patients 10 to 20 times higher than in the general population [44]. However, the usefulness of creatinine or eGFR for predicting cardiovascular outcomes in CKD patients is still controversial. In our study, serum creatinine levels, eGFR, and diabetes were predictive of cardiovascular events and death. According to the latest meta-analysis by Matsushita et al. [45], creatinine-based eGFR and albuminuria independently improved the prognosis of incident cardiovascular outcomes. Another option is to measure GFR using other filtration markers, such as cystatin C (eGFRcys). Recently, Lees et al. [46] demonstrated that eGFRcys is more strongly associated with CVD and mortality than creatinine-based measurement. Therefore, eGFR should be considered for cardiovascular prediction, especially when it is already assessed for clinical purposes. The latest studies also show that diabetes is associated with a risk for CV events in HD [47], which is in line with our results.

To sum up, there was no difference in serum levels of α-Klotho between HD patients and healthy people, though, HD patients had increased serum IL-6 and IL-8. Significant positive correlations existed between the concentration of α-Klotho and the serum concentration of IL-12p70, IL-10, and IL-1β. However, in a multivariable linear regression analysis, only patients’ age was associated independently with α-Klotho level. Serum α-Klotho was not associated with higher mortality risk in HD patients.

In summary, these associations, which we demonstrated in our study, draw attention to the potential relationship between α-Klotho levels, the inflammation status, and mortality of HD patients. Based on the present state of knowledge, serum α-Klotho measurement does not appear to be helpful in patients with ESRD to predict mortality. However, our cross-sectional study has several limitations, including the relatively small size of the study, the Polish origin of the dialysis population, and a small number of deaths during the observation period. Therefore, these results may not be generalized. Consequently, further studies are necessary to clarify the relationship between α-Klotho, inflammation, and mortality in hemodialysis patients. In addition, the development of a standardized assay is essential to measure circulating α-Klotho and to demonstrate its clinical relevance.

## Figures and Tables

**Table 1 jcm-11-06518-t001:** Comparison of basic clinical parameters and serum concentrations of HD patients and healthy people.

	Healthy People (*n* = 19)	HD Patients (*n* = 67)	*p* Value
Age (years)	70 (37, 87)	69 (36, 89)	0.613
HD time (months)	n.a.	28 (0.5, 169)	n.a.
Creatinine (mg/dL)	n.a.	6.3 (2.71, 10.38)	n.a.
eGFR (min/mL/1.73 m^2^)	n.a.	7 (4, 22)	n.a.
Kt/V	n.a.	1.5 (0.74, 2.37)	n.a.
PTH (pg/mL)	n.a.	356 (44.5, 4276)	n.a.
Ca (mg/dL)	n.a.	8.7 (6.5, 10.8)	n.a.
Pi (mg/dL)	n.a.	5.5 (2.4, 10.5)	n.a.
IL-12p70 (pg/mL)	2.7 (1.05, 12.74)	2.45 (0, 108.95)	0.737
TNF (pg/mL)	2.59 (0.84, 5.22)	2.5 (0, 127.65)	0.568
IL-10 (pg/mL)	2.65 (0.72, 6.15)	2.8 (1.4, 21.88)	0.104
IL-6 (pg/mL)	**4.08 (2.23, 22.57)**	**10.3 (3.45, 805.45)**	**<0.001**
IL-1β (pg/mL)	1.83 (0, 3.33)	1.75 (0, 18.24)	0.844
IL-8 (pg/mL)	**13.56 (9.59, 911.66)**	**38.8 (10.4, 5000)**	**0.001**
α-Klotho (pg/mL)	8.74 (1.33, 60.82)	6.96 (0.45, 127.1)	0.451

Data are presented as median with minimum and maximum. The results in bold are statistically significant, Mann-Whitney U test, *p* < 0.05. eGFR—estimated glomerular filtration rate, PTH—parathormone, Ca—calcium, Pi—phosphorus, n.a.—not applicable.

**Table 2 jcm-11-06518-t002:** Correlations between serum concentrations of α-Klotho, patients’ clinical parameters, and serum cytokines.

	Age	HD Time	Cr	eGFR	Kt/V	PTH	Ca	Pi	IL-12	TNF	IL-10	IL-6	IL-1	IL-8	α-KL
Age	1.000	−0.186	**−0.358**	0.192	−0.194	**−0.253**	0.106	**−0.330**	−0.138	−0.076	−0.178	0.047	−0.048	0.053	**−0.268**
HD time		1.000	0.178	**−0.286**	**0.415**	**0.449**	0.036	−0.106	−0.003	0.042	0.045	0.028	−0.004	**−0.278**	−0.004
Cr			1.000	**−0.668**	**0.294**	0.143	−0.223	**0.408**	−0.122	−0.064	−0.119	−0.004	−0.080	0.189	−0.108
eGFR				1.000	**−0.353**	**−0.327**	0.108	**−0.415**	0.125	−0.053	0.094	0.121	−0.069	−0.184	0.075
Kt/V					1.000	0.228	−0.007	0.064	0.014	0.083	−0.088	−0.125	−0.050	−0.020	−0.138
PTH						1.000	−0.146	**0.242**	−0.017	0.165	−0.073	0.024	−0.026	0.081	**−0.290**
Ca							1.000	−0.217	0.113	0.018	−0.038	−0.162	0.140	−0.143	0.117
Pi								1.000	0.026	0.207	0.034	0.074	0.158	0.186	−0.025
IL-12									1.000	**0.384**	**0.424**	0.235	**0.421**	−0.028	**0.250**
TNF										1.000	**0.286**	0.134	0.179	−0.018	−0.191
IL-10											1.000	**0.386**	**0.546**	−0.022	**0.333**
IL-6												1.000	0.195	**0.364**	0.090
IL-1													1.000	−0.110	**0.436**
IL-8														1.000	−0.123
α-KL															1.000

The results in bold are statistically significant, bivariate Spearman Rank Correlation test, *p* < 0.05. Cr—creatinine, α-KL—α-Klotho.

**Table 3 jcm-11-06518-t003:** Multivariable linear regression analyses for α-Klotho.

Variable	Estimate	Standard Error	95% CI	*p* Value
Age	**−0.551**	**0.271**	**−1.09 to −0.00817**	**0.047**
PTH	−0.00779	0.00491	−0.0176 to 0.00202	0.118
IL-12	−0.0647	0.167	−0.399 to 0.270	0.700
IL-10	−0.545	0.997	−2.54 to 1.45	0.587
IL-1	1.71	1.15	−0.601 to 4.01	0.144

The results in bold are statistically significant (*p* < 0.05). CI—confidence interval.

**Table 4 jcm-11-06518-t004:** Cox regression analyses of cardiovascular events, cardiovascular mortality, and all-cause mortality.

	Cardiovascular Events		Cardiovascular Mortality		All-Cause Mortality	
	Hazard Ratio (95% CI)	*p* Value	Hazard Ratio (95% CI)	*p* Value	Hazard Ratio (95% CI)	*p* Value
Sex [male]	0.685 (0.335–1.44)	0.305	1.73 (0.407–11.7)	0.500	1.00 (0.339–3.30)	0.999
Age (years)	**1.03 (1.00–1.07)**	**0.038**	1.02 (0.971–1.09)	0.396	1.04 (0.990–1.00)	0.144
HD time (months)	1.00 (0.994–1.01)	0.610	1.00 (0.983–1.01)	0.901	0.99 (0.984–1.01)	0.909
Creatinine (mg/dL)	**0.726 (0.575–0.902)**	**0.005**	**0.517 (0.313–0.807)**	**0.006**	**0.577 (0.392–0.816)**	**0.003**
eGFR (min/mL/1.73 m^2^)	1.04 (0.912–1.16)	0.568	**1.25 (1.07–1.46)**	**0.003**	**1.20 (1.05–1.36)**	**0.004**
Kt/V	0.560 (0.162–1.92)	0.356	0.888 (0.097–8.21)	0.916	0.629 (0.105–3.72)	0.610
PTH (pg/mL)	1.00 (1.00–1.00)	0.347	1.00 (0.999–1.00)	0.463	1.00 (0.999–1.00)	0.711
Ca (mg/dL)	1.27 (0.752–2.23)	0.389	2.24 (0.814–6.00)	0.119	1.62 (0.731–3.56)	0.240
Pi (mg/dL)	0.899 (0.715–1.13)	0.357	0.848 (0.544–1.29)	0.452	0.741 (0.512–1.05)	0.101
IL-12p70 (pg/mL)	0.987 (0.959–1.00)	0.230	0.994 (0.944–1.02)	0.712	0.989 (0.936–1.01)	0.512
TNF (pg/mL)	0.968 (0.882–1.00)	0.303	0.948 (0.699–1.01)	0.566	0.918 (0.680–1.01)	0.415
IL-10 (pg/mL)	1.01 (0.909–1.08)	0.882	1.04 (0.860–1.16)	0.579	1.01 (0.835–1.12)	0.915
IL-6 (pg/mL)	0.997 (0.985–1.00)	0.388	0.999 (0.978–1.00)	0.723	0.999 (0.982–1.00)	0.681
IL-1β (pg/mL)	0.976 (0.855–1.08)	0.677	0.956 (0.699–1.14)	0.708	0.884 (0.646–1.07)	0.339
IL-8 (pg/mL)	0.999 (0.995–1.00)	0.281	0.999 (0.995–1.00)	0.573	0.999 (0.996–1.00)	0.505
α-Klotho (pg/mL)	0.997 (0.983–1.01)	0.664	0.996 (0.964–1.02)	0.759	0.991 (0.960–1.01)	0.438
Diabetes [yes]	**2.73 (1.31–6.11)**	**0.009**	3.86 (0.900–26.7)	0.100	2.45 (0.802–9.10)	0.138

The results in bold are statistically significant (*p* < 0.05).

## Data Availability

The data presented in this study are available on request from the corresponding author.

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
