# Peer review of "Serum Levels of α-Klotho, Inflammation-Related Cytokines, and Mortality in Hemodialysis Patients"

_jcm, 2022, doi:10.3390/jcm11216518_

Round 1
Reviewer 1 Report
The aim of the study was to show clinical data on the levels of klotho and inflammatory cytokines in the serum of HD patients. The study is a typical 'observatory' with no hypothesis or mechanism suggestion. However, it shows relevant clinical data based on the well-designed and well-performed project. The conclusions are fully supported, the methods are correct, and the manuscript is well written. I have only a few major issues:
- in Table 2, the Authors sometimes use ',' and sometimes '.' in decimal notation.
- also, the discussion should be supplemented with recent literature - there are several new papers on the role of klotho in inflammation
- nice graphics summarizing the observations should be added in the discussion section
Author Response
Answers to the first Reviewer. Corrections are marked in yellow.
“I have only a few major issues:
- in Table 2, the Authors sometimes use ‘,’ and sometimes ‘.’ in decimal notation,”
We corrected the punctuation in Table 2 as suggested. Thank you for pointing it out.
“- also, the discussion should be supplemented with recent literature - there are several new papers on the role of klotho in inflammation,”
We added some new literature to the discussion. However, in many cases, the latest papers did not mention inflammation-related cytokines or were carried out in a completely different group of patients, e.g., alcoholics or patients with metabolic syndrome. We included one of the latest studies, for example: (1) Martín-Núñez E et al. Aging 2020 (https://doi.org/10.18632/aging.102734), (2) Gamrot Z et al. Endokrynol. Pol. 2021(https://doi.org/10.5603/ep.a2021.0082), (3) Wei H et al. BMC Nephrol. 2019 (https://doi.org/10.1186/s12882-019-1232-2), (4) Oh HJ et al. Perit. Dial. Int. 2015 (https://doi.org/10.3747/pdi.2013.00150), (5) Lee HJ et al. Aging 2022 (https://doi.org/10.18632/aging.203987)m (6) Wu SE et al. Ann. Med. 2022 (https://doi.org/10.1080/07853890.2022.2077428), (7) Mytych J et al. Mol. Cell Endocrinol. 2018 (https://doi.org/10.1016/j.mce.2017.05.003), (8) Junho CVC et al. Biomed Pharmacother. 2022 (https://doi.org/10.1016/j.biopha.2022.113515).
“- nice graphics summarizing the observations should be added in the discussion section.”
In answer to a proposal for graphics summarizing our results, we prepared graphical abstract.
Author Response
Answers to the second Reviewer. Corrections in the text are marked in green.
“My comments and concerns were listed below. (My comments are in bold)
- In the materials and methods section, Sub sections 2.1, 2.2 &2.3
Does freezing and thawing the samples have any effect on the levels of detection of α-Klotho and Cytokine levels? I recommend to provide some literature review on this.”
We added the literature in section 2.1. about preparing samples. Unfortunately, there is no option to analyze fresh serum samples in human cohort studies. To our knowledge, serum sample storage at -80°C is the best form of storing samples for clinical studies (https://www.mayocliniclabs.com/test-catalog/overview/75139#Specimen). Also, the manufacturer of the CBA Human Inflammatory Cytokine Kit states that the test is optimized for the analysis of specific proteins in tissue culture supernatants, EDTA-treated plasma, and serum samples.
One-time freezing and thawing do not affect serum cytokine levels as long as samples are stored at -80°C (Graham C et al. J Transl Med. 2017 Mar 2;15(1):53). In addition, some authors show that even a repeated freezing-thawing cycle does not influence cytokine levels. However, we only thaw samples once in our lab and store them at -80°C. We also collected blood into gel-separator serum tubes, and samples were frozen immediately after centrifugation (Verberk IM et al. Biomark Med. 2021 Aug;15(12):987-997). We also measured cytokines in duplicates to validate measurements. However, it should be emphasized that the authors studying the influence of sample processing use different methods for evaluating serum cytokines. So far, we have found only one article comparing different methods that results obtained with Luminex and CBA kits correlated with ELISA (Richens JL et al. J Biomol Screen. 2010 Jun;15(5):562-8).
As to serum Klotho, we could not find any studies comparing ELISA results in different human samples. Therefore, we used the same sample processing procedure. Also, similar sample processing was used by other authors studying serum Klotho in ESRD patients.
“2. In discussion section I recommend to add an initial paragraph summarizing the findings.”
We added a paragraph summarizing the findings in the Discussion. However, we attached it at the end of the section before the final conclusions. We also added graphics summarizing our results.
Round 2
Reviewer 1 Report
The authors have addressed all issues and the manuscript deserves to be published.